# A bibliometric analysis of geographic disparities in the authorship of leading medical journals

Oscar Brück [1✉]

**Abstract**

**Background** It has previously been reported that authors from developing countries are underrepresented in medical journals. Here, we aimed to build a comprehensive landscape of the geographical representation in medical research publications.

**Methods** We collected bibliometric data of original research articles ($n = 10{,}558$) published between 2010 and 2019 in five leading medical journals and geolocated these by the institute of the corresponding authors. We introduced two simple metrics, the International Research Impact and the Domestic Self-Citation Index, to assess publishing and citing patterns by cities and countries.

**Results** We show that only 32 countries published more than 10 publications in 10 years equaling 98.9% of all publications. English-speaking countries USA (48.2%), UK (15.9%), Canada (5.3%), and Australia (3.2%) are most represented, but with a declining trend in recent years. When normalized to citation count, 9/32 countries published $\geq 10\%$ more than expected. In total, 85.7% of the publication excess originate from the USA and UK. We demonstrate similar geographical bias at the municipal level. Finally, we discover that journals more commonly publish studies from the country in which the journal is based and authors are more likely to cite work from their own country.

**Conclusions** The study reveals Anglocentric dominance, domestic preference, but increased geographical representation in recent years in medical publishing. Similar audits could mitigate possible national and regional disparities in any academic field.

**Plain language summary**

Geographical representation in authorships of research articles is insufficiently understood. We analyzed data from over 10,000 research articles published between 2010–2019 in top medical journals. Anglocentric countries (USA, UK, Canada, and Australia) accounted for most publications, but their proportion has recently declined. When considering citations, i.e. formal references connecting new findings to observations from previously published articles, 1/3 of the studied countries published ≥10% more articles than expected. When publishing and citing articles, journals and researchers tended to favor publications from their own countries. While some improvement in geographical representation has occurred, our findings expose an Anglocentric bias and national preference, which might bias medical publishing. The approach used in this study may be used in future efforts to monitor geographical representation in publication authorships.

[1] Hematoscope Lab, Comprehensive Cancer Center & Center of Diagnostics, Helsinki University Hospital, Helsinki, Finland & Department of Oncology, University of Helsinki, Helsinki, Finland. ✉email: oscar.bruck@hus.fi

Diversity in academic publishing is critical to mitigate disparities in health care and promote fair and comprehensive dissemination of medical knowledge. By signing the "Joint commitment for action on inclusion and diversity in publishing" more than 50 publishers representing 15,000 scientific journals will collect data on gender and ethnical representation to identify bias in the editorial and review processes[1]. Despite the importance of prospective bibliometric data collection, it is unclear who will have access to analyze the resource. In addition, temporal trends and the current state of geographical inclusion might not be ideally captured.

Previously, the underrepresentation of developing countries has been described in medical journals[2], but to our knowledge, no comprehensive landscape has been composed on the geographical representation in medical research. Information on submitted and rejected manuscripts is not available. Therefore, retrospective studies need to examine accepted publications. Publication metadata stored in comprehensive journal indexing databases such as the Web of Science (WoS), Scopus, PubMed, and Google Scholar could better interrogate publishing and citing patterns based on the geographical origin of the conducted research. Medical journals with high journal impact factors (JIF) are particularly interesting to study. First, the editorial and review processes of these journals are exigent to guarantee the publication of cutting-edge research. Therefore, we hypothesized that prominent medical journals should not differ by the geographical representation of their publications. Second, JIFs reflect the averaged citation patterns of one journal at a certain time period. We hypothesized that citation rates of articles published in prominent medical journals should not differ depending on the geographical location of the conducted research.

In this retrospective study, we examine possible bias related to the geographical location of the conducted research to the number of publications and citations. We focus on the five most prominent medical journals selected based on their journal impact factor (JIF) in 2022. While not all articles published in these journals represent scientific breakthroughs, JIFs reflect citation patterns and interest in the research community.

This study demonstrates that medical publishing is marked by Anglocentric dominance. Moreover, journals tend to publish more studies from the country in which the journal is based, and authors are more likely to cite work from their own country. However, geographical representation has gradually increased in recent years. In summary, understanding the geographical landscape of scientific publishing in leading medical journals in particular, could help us to identify disparities at the international and even national level.

## Methods

**Data collection.** We selected the five journals publishing mainly original articles in all fields of medicine and ranked highest in the Journal Citation Reports 2022 JIF to exclude possible bias related to specialized medical fields. These included *New England Journal of Medicine* (NEJM), *Nature Medicine* (NatMed), *Journal of the American Medical Association* (JAMA), *The BMJ*, and *Lancet*. We included all original articles ($n = 10,558$) published in 2010–2019. More recent publications were not included due to brief follow-up and to avoid bias related to COVID-19 pandemic. Using the WoS database provided by Clarivate Plc and the query "(((SO = (NATURE MEDICINE OR LANCET OR NEW ENGLAND JOURNAL OF MEDICINE OR JAMA JOURNAL OF THE AMERICAN MEDICAL ASSOCIATION OR BMJ BRITISH MEDICAL JOURNAL)) AND DT = (Article)) AND PY = (2010–2019))", we downloaded publication metadata including author names, page length of the article, institute

address of the corresponding author, and total number of citations. We could also query international citing patterns based on the digital object identifiers of the 10,558 original articles. National population data were downloaded from the World Bank database    https://data.worldbank.org/indicator/SP.POP.TOTL, and municipal data from the world.cities dataset of the utils R package.

**Preprocessing variables of interest.** There are multiple measures to assess the impact of an article. Here, we compared articles by their average citation count per year reflecting their scientific interest and influence. We text-mined the number of authors, by calculating the frequency of the semicolon ";" delimiter between author names and added 1. To geolocate the primary institutes where the research has been performed, we identified the latitude and longitude coordinates of the address(es) of the corresponding author(s) with the ggmap library employing Google's Geocoding API. For non-successful matches, we geolocated only the city and country of the address. In total, only two institutes could not be geolocated due to insufficient information.

**The International Research Impact (IRI) index.** This statistical measure can be calculated by dividing the total number of publications (10-fold logarithm transformed) from any given city by their cumulative citations (10-fold logarithm transformed). While IRI is calculated at the municipal level, we calculated the median IRI scores also by countries to permit comparisons at the international level.

**The Domestic Self-Citation Index (DSCI).** To study citation patterns as a source of geographical bias, we examined in more detail the geolocation of publication citations. For this purpose, we introduce DSCI:

$$\frac{\#\,Citations\ from\ country\ \alpha\ to\ country\ \alpha}{\#\,Total\ citations\ from\ country\ \alpha} \quad (1)$$

$$\frac{\#\,Citations\ to\ country\ \alpha\ from\ all\ countries\ except\ country\ \alpha}{\#\,Citations\ in\ total\ to\ all\ countries\ except\ country\ \alpha\ from\ country\ \alpha} \quad (2)$$

where "# Citations" represents the number of citations and α any country of interest. The first part (1) of the equation reflects the proportion of national self-citations (%) of all citations. The second part (2) normalizes the first part by observing the proportion (%) of citations accorded to a country by all other countries.

**Statistical analysis.** We performed comparisons of two continuous variables with the Wilcoxon rank-sum test (unpaired, two-tailed) and three or more continuous variables with the Kruskal–Wallis test. We adjusted p values with the Benjamini–Hochberg correction. We performed hierarchical clustering with Euclidean distance metrics and the Ward.d2 method. To compare two linear regression slopes, we tested the T-test significance of the interaction term. We conducted statistical analyses and visualizations with R 3.5.1. using base, tidyverse, fastDummies, maps, reshape2, ggmap, data.table, countrycode, ggpubr, ggrepel, rstatix, ggdendro and dendextend libraries.

**Reporting summary.** Further information on research design is available in the Nature Portfolio Reporting Summary linked to this article.

## Results

**National publication productivity does not imply high citation frequency.** Of 10,558 original articles published between 2010–2019 in *NEJM* ($n = 2966$; 28.1%), *JAMA* ($n = 1833$; 17.4%), *NatMed* ($n = 1577$; 14.9%), *Lancet* ($n = 2462$; 23.3%), and *BMJ* ($n = 1720$; 16.3%), corresponding authors were affiliated to institutes from 77 countries. Each article included 1.2 mean affiliations [range 1–15] equaling 10,732 total unique entries, of which 100.0% ($n = 10,730$) could be geolocated. When focusing on countries with ≥10 original articles in 10 years in any of these five medical journals, only 32 countries were included representing 98.9% ($n = 10,613$) affiliations highlighting the geographical exclusivity of medical research.

Of these, English-speaking countries were overrepresented with almost 3/4 of publications from the United States of America (USA, 48.2%), the United Kingdom (UK, 15.9%), Canada (5.3%), and Australia (3.2%, Table 1). National population-normalized publication frequencies correlated with total publication number (corr 0.69 $p < 0.001$, Spearman test). Following population normalization, the most represented countries included Denmark (31.1 articles per million people), UK (25.1), and Switzerland (24.8, Table 1).

To evaluate international research impact, we made two assumptions. First, we reasoned that the proportion of citations by publications should not differ by the country where the research has been conducted as the publication acceptance criteria should be equal. Second, research institutions might share several geolocations. Therefore, by aggregating publication and citation counts at the municipal level would permit a more reliable comparison of geographical bias. For this purpose, we introduce the International Research Impact Index (IRI, see Methods).

When examining the IRI by countries, publication count explained 96.5% of total citations corresponding to an excellent statistical correspondence (Fig. 1a). However, we also observed unexpected discrepancies. We identified 19/32 countries publishing ≥10% more (publication excess) or less (publication deficit) articles than was expected based on their accumulated citations. In terms of publication count, most excess articles were published by corresponding authors affiliated with an organization located in the USA ($n = 1174$ excess articles corresponding to 23.0% of all articles from the USA) and UK ($n = 410$ or 24.3% more articles than expected). Together, these accounted for 85.7% of all articles published in excess. When normalizing the absolute number of excess/deficit articles to each country's total article count, Spain published 78.7% ($n = 76$) and Brazil 62.1% ($n = 26$) fewer articles than expected.

Publications with a corresponding author from Saudi Arabia were associated with the highest median yearly citations (49.4) per publication, which was partly explained by their emphasis on the documentation of the Middle East Respiratory Syndrome (7 out of 11 total publications from Saudi Arabia). Other countries with the most elevated median yearly citation per publication included Belgium (40.2) and China (39.6). Lowest median citations were associated with publications geolocated in South Asia (Pakistan 11.6, Bangladesh 14.6, India 21.3) and Scandinavia (Norway 19.7, Denmark 21.2, Sweden 21.3).

When extrapolating to the continental level, publication exclusivity was even more pronounced with 89.5% of all articles originating either from the Americas (53.4%) or Europe (36.1%, Fig. 1b). However, 5.2% of publications originating from Asian countries were associated with 15.6% higher citation frequencies compared to the median (27.5 citations/article; Fig. 1b, c).

**Table 1 Publication metrics by country.**

| Country | # Total articles | # Articles/ population | # Total Citations $(10^3)$ | # Citations/ year | # Citations/article/ populaton/year | # Excess articles | % Excess articles |
|---|---|---|---|---|---|---|---|
| USA | 5111 | 15.4 | 305.1 | 28.0 | 0.1 | 1174.5 | 23.0 |
| UK | 1691 | 25.1 | 85.6 | 24.4 | 0.4 | 410.5 | 24.3 |
| Canada | 560 | 14.6 | 30.6 | 28.4 | 0.7 | 43.8 | 7.8 |
| France | 386 | 5.7 | 24.1 | 33.7 | 0.5 | −32.1 | −8.3 |
| Germany | 376 | 4.5 | 21.3 | 36.0 | 0.4 | 1.3 | 0.3 |
| Netherlands | 374 | 21.3 | 17.7 | 25.9 | 1.5 | 55.6 | 14.9 |
| Australia | 339 | 13.2 | 20.4 | 28.9 | 1.1 | −22.1 | −6.5 |
| China | 225 | 0.2 | 15.9 | 39.6 | 0.0 | −64.5 | −28.7 |
| Switzerland | 216 | 24.8 | 10.2 | 29.9 | 3.4 | 20.4 | 9.4 |
| Sweden | 199 | 19.1 | 7.3 | 21.2 | 2.0 | 53.6 | 26.9 |
| Denmark | 182 | 31.1 | 6.6 | 21.2 | 3.6 | 48.5 | 26.6 |
| Italy | 133 | 2.3 | 6.7 | 35.3 | 0.6 | −1.7 | −1.3 |
| Japan | 112 | 0.9 | 6.0 | 26.9 | 0.2 | −9.9 | −8.8 |
| Spain | 97 | 2.0 | 8.9 | 39.6 | 0.8 | −76.3 | −78.7 |
| Belgium | 82 | 7.1 | 5.1 | 40.2 | 3.5 | −24.1 | −29.4 |
| Norway | 62 | 11.5 | 1.9 | 19.6 | 3.6 | 17.0 | 27.5 |
| South Africa | 58 | 1.0 | 2.4 | 25.5 | 0.4 | 4.3 | 7.3 |
| New Zealand | 53 | 10.3 | 2.2 | 27.8 | 5.4 | 1.8 | 3.4 |
| Israel | 47 | 5.0 | 2.0 | 22.0 | 2.3 | −0.3 | −0.5 |
| Brazil | 42 | 0.2 | 3.1 | 37.1 | 0.2 | −26.1 | −62.1 |
| Finland | 40 | 7.2 | 1.7 | 26.1 | 4.7 | 0.3 | 0.8 |
| Austria | 36 | 4.0 | 1.6 | 29.8 | 3.3 | −1.9 | −5.2 |
| South Korea | 34 | 0.7 | 1.7 | 39.1 | 0.8 | −5.2 | −15.3 |
| India | 34 | 0.0 | 1.3 | 21.3 | 0.0 | 2.8 | 8.4 |
| Singapore | 23 | 4.2 | 1.2 | 23.2 | 4.3 | −5.9 | −25.6 |
| Ireland | 17 | 3.4 | 0.7 | 29.5 | 5.9 | −1.8 | −10.7 |
| Thailand | 17 | 0.2 | 0.9 | 26.7 | 0.4 | −6.9 | −40.6 |
| Kenya | 17 | 0.3 | 0.5 | 23.7 | 0.4 | 4.5 | 26.7 |
| Greece | 14 | 1.3 | 0.7 | 31.9 | 3.0 | −5.1 | −36.2 |
| Pakistan | 13 | 0.1 | 0.4 | 11.6 | 0.1 | 2.4 | 18.5 |
| Saudi Arabia | 12 | 0.3 | 0.6 | 49.4 | 1.4 | −4.9 | −40.6 |
| Bangladesh | 11 | 0.1 | 0.2 | 14.6 | 0.1 | 6.0 | 54.3 |

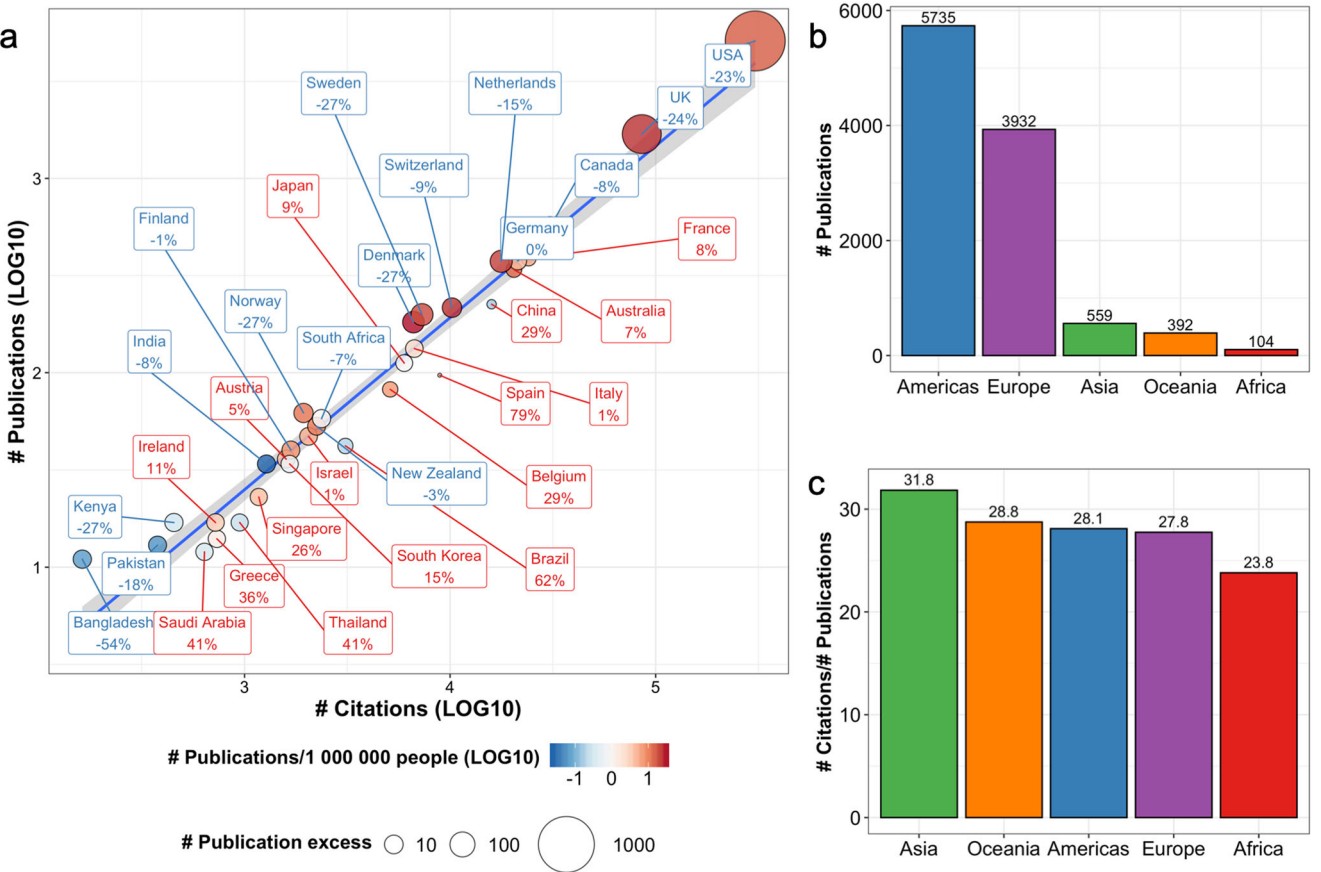

**Fig. 1 Medical publishing at the national level. a** The International Research Impact Index at the national level has been visualized with a linear regression of the number of citations (LOG10-transformed) and publications (LOG10-transformed) of the top 32 most productive countries. The percentage and font color indicate the proportion of excess (positive number and red font) or deficient (negative number and blue font) publications compared to the predicted number based on the citation/publication count. The shaded area represents the 95% confidence interval of the regression model. **b** Bar plots illustrating the number of publications and (**c**) citations per publication by continent.

We then studied how the number of publications per country has changed during the study time period. For this purpose, we fitted linear regression curves for the publication count (Supplementary Fig. 1a) for each country by time. Yearly publishing productivity in the studied journals decreased steadily in the USA (8.0 fewer articles per year), UK (5.7), Canada (1.7), and Italy (1.2). However, the number of publications originating from China (2.5) and Israel (0.89) rose, but with a lower inclination. The proportion of citations per publication increased from 2010 to 2019 in countries with high publication count, with the exception of Japan where the trend was not as evident (Supplementary Fig. 1b).

**Publications from elite research institutes tend to accumulate fewer citations per article.** The international publication selection bias was even more pronounced when studying the regional location of the corresponding authors' institutes (Fig. 2a). Distinct publication hot spots were identified in Northeastern USA, Central Europe, and the UK. On the contrary, the number of citations per article did not replicate similar geographical patterns (Fig. 2a). The 10 most represented cities in terms of publication count covered 36.5% of all publications and were all located in the Anglosphere: USA ($n = 6$), UK ($n = 3$) and Canada ($n = 1$). The three cities with over 300 publications in 2010–2019 were Boston, USA ($n = 1209$, 12.2% of all publications), London, UK ($n = 648$, 6.5%), and New York, USA ($n = 378$, 3.8%).

Next, we examined IRIs by including only cities with ≥10 publications in 10 years retaining 94.0% of the number of publications ($n = 9327$). Publication count explained 83.6% of the variability in total accumulated citations (Fig. 2b). Multiple metropolitan cities known for their established research institutes were overrepresented in terms of publication frequencies (Fig. 2b). Unexpectedly, the residuals of the linear regression model increased by publication number (corr 0.37, $p < 0.001$). In consequence, 21 out of the 25 most productive cities accumulated fewer citations than predicted with linear regression. Cities with the highest absolute publication excess included Boston, USA ($n = 728$ excess articles), London, UK ($n = 285$), and Oxford, UK ($n = 76$). On the contrary, cities with the highest publication deficit included Seattle, USA ($n = 76$), Houston, USA ($n = 27$), and Los Angeles, USA ($n = 6$). However, capital cities generally associated with elevated population density and administrative and business activity, did not accumulate more citations compared to non-capital cities (Supplementary Fig. 2).

While the publication count decreased in the USA and UK during the 10-year follow-up, similar patterns were not as evident when examining the cities publishing most articles (Supplementary Fig. 3a). Some cities published gradually more (New York, Stanford, Philadelphia), some less (London, Baltimore, Ann Arbor), but for most the direction was inconclusive. At the national level, the yearly citations-per-article measure increased on average (Supplementary Fig. 3b).

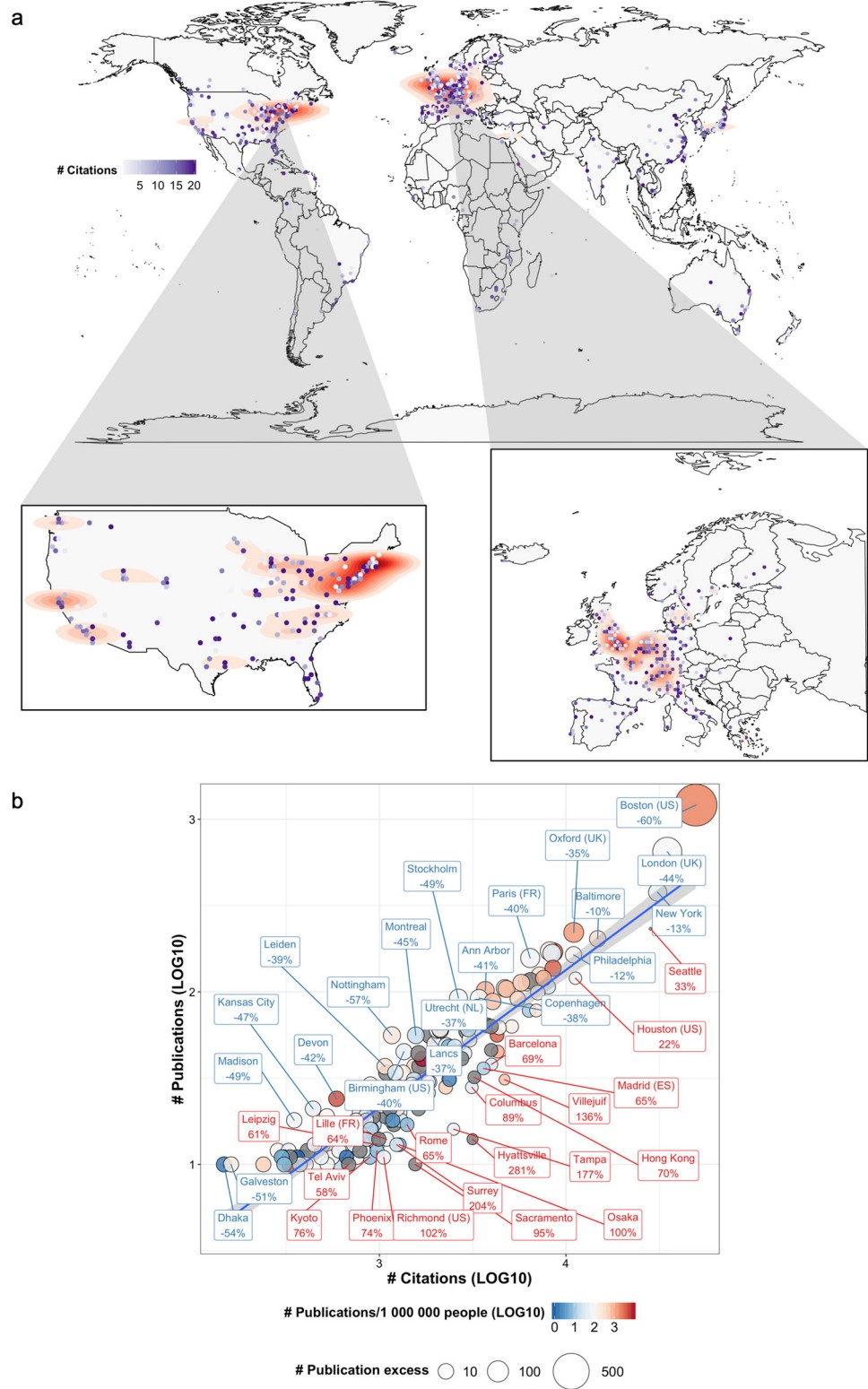

**Fig. 2 Medical publishing at the municipal level. a** A world map with density contours indicating hot spots based on the number of publications. Blue-colored points reflect the average number of citations in that area. **b** The International Research Impact Index at the municipal level has been visualized with a linear regression of the number of citations (LOG10-transformed) and publications (LOG10-transformed) by cities. The percentage and font color indicate the proportion of excess (positive number and city name in red) or deficient number of publications compared to the predicted number based on the citation/publication count. The shaded area represents the 95% confidence interval of the regression model. Country name abbreviations are added for city names, which are found in more than one country.

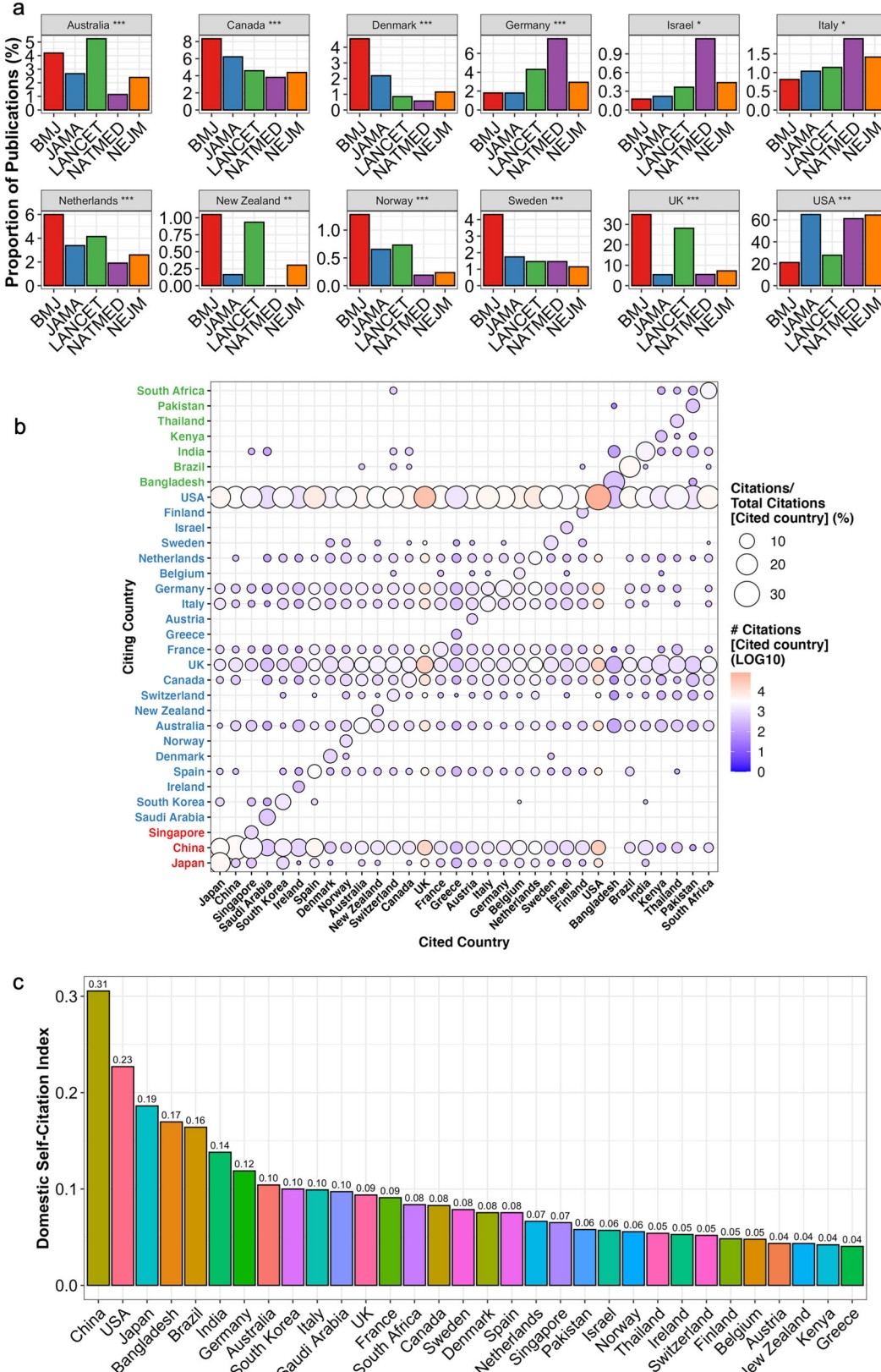

**Institute nationality affects both journalistic and citation patterns**. Given overlap in the scientific scope of the studied five medical journals, we reasoned that the geographical coverage of publications should not differ. However, we observed significantly more publications from English-speaking countries (UK, New Zealand, and Australia) in both UK-based *BMJ* and *Lancet* journals (Fig. 3a). Publications from Scandinavian countries (Norway, Sweden, and Denmark), Netherlands and Canada were overrepresented in *BMJ*. Publications from Germany, Israel, and Italy were most common in *NatMed*, which is part of the

**Fig. 3 National journal and citation fingerprints. a** Bar plot illustrating the proportion of publications in medical journals originating from different countries. Only significant results of the Kruskal–Wallis analysis are plotted. **b** Balloon plot illustrating the international citation patterns of articles published in leading medical journals. The balloon size reflects how much citations from a Citing country (*y*-axis) constitute of a Cited country's total received citations (*x*-axis). The balloon color reflects the absolute number of citations a Citing country (*y*-axis) accords a Cited Country (*x*-axis). Citing countries have been grouped by hierarchical clustering into three subgroups (*y*-axis font color). **c** Bar plot illustrating the same data as in the previous plot but proportioned to how much other countries have cited any given country in relation to how much they have cited all other countries. See Methods for the definition of the Domestic Self-Citation Index.

British–German Springer publishing group. Publications from institutes located in the USA were overrepresented in *NatMed* and US-based journals *JAMA* and *NEJM*.

To conclude, we audited national citation patterns by (1) comparing citation counts received by any country, (2) comparing how frequently researchers cite publications originating from the same country, and (3) identifying possible international citation patterns.

First, countries with the most publications were also the most active to cite as expected (Fig. 3b, see rows of balloons). Of the 32 studied countries, 30 were most commonly cited by a research group from USA. The last two countries (China and Singapore) were cited more commonly by publications with a corresponding author based in China.

Second, domestic preference was evident when evaluating national citation patterns (Fig. 3b, see diagonal line of balloons). Researchers affiliated with an institute in USA, UK, or South Africa demonstrated the highest absolute domestic self-citation (Supplementary Fig. 4). However, this result is biased as it does not observe whether researchers from other countries would also cite publications from these countries. To correctly measure the domestic self-citation of a country of interest, we introduce DSCI, which normalizes self-citations with the proportion of citations from all other countries (see Methods "Domestic Self-Citation Index (DSCI)" and robustness tests and data in Supplementary Table 1 and Supplementary Fig. 5).

The highest DSCI associated with China, USA, and Japan, and the lowest with Austria, New Zealand, Kenya, and Greece (Fig. 3c). National DSCI correlated with their total accumulated citation count (corr 0.70, $p < 0.001$) and publication count (corr 0.46, $p = 0.015$). However, DSCI did not correlate with the absolute count (corr 0.032, $p =$ ns) or proportion of excess articles (corr $-0.060$, $p =$ ns). Domestic self-citation accounted for 74.5% of the total national citation count (corr 0.86, $p < 0.001$) indicating the serious bias related to citation count for evaluating manuscript impact.

Lastly, we examined international citation patterns. We observed that national citation patterns formed distinct clusters (Fig. 3b, colored row names). The largest cluster was composed of the Anglosphere, European countries, South Korea, and Saudi Arabia. The second largest cluster included developing countries whereas China, Japan, and Singapore formed the third cluster. In summary, similar geographical location, development index and health challenges accounted for citation trends at the international level.

## Discussion

Available bibliometric data can reveal important information on geographical representation in medical research and its temporal dynamics. Here, we presented geographical publication disparity in five leading medical journals between 2010 and 2019.

First, we described the overrepresentation of distinct countries and research organizations. Almost 3/4 of all publications with a corresponding author originated from an institute in USA, UK, Canada, or Australia indicating medical publishing is Anglo-centric. Similar findings have been reported both in leading

medical journals between 1971 and 2005 and in the field of pathology between 2000 and 2006 suggesting both an interdisciplinary and long-lasting phenomenon[3, 4]. We showed that the number of articles from institutes in USA, UK, and Canada was decreasing, while publications from China and Israel were more frequent reflecting a gradual transition in international representation.

Second, by comparing accumulated citations per article (or IRIs) our analysis highlighted an undocumented discrepancy favoring notably highly productive institutes and countries. The results indicate that IRI is an objective and simple measure to study geographical diversity both in medical or interdisciplinary domains and its temporal evolution.

Multiple factors could be involved. For instance, previous studies have indicated a pronounced correlation between national gross domestic product and publishing in top medical journals[5, 6]. Our data indicated higher proportions of articles from UK, Australia, Canada, and New Zealand in UK-based journals (*BMJ* and *Lancet*), while corresponding authors from USA were overrepresented in American journals (*JAMA* and *NEJM*). UK-affiliated research published in UK-based journals *BMJ* and *Lancet* (mean 30.8%) was almost 5-fold compared to the average proportion in *NEJM*, *NatMed* and *JAMA* (6.3%). Importantly, journals such as *Lancet* advice authors to consider diversity when inviting coauthors[7]. Yet, further comparative studies on submission rates are required to better understand the reasons and to exclude domestic preference at the review or editorial level, which could undermine public trust in scientific publishing and possibly find new journals from underrepresented countries.

Our data indicated that less-publishing countries and institutes tended to accumulate higher citations, which could reflect both interest and ultimately elevate further JIFs. Diversity and inclusion of developing countries would benefit the scientific community as demonstrated during the SARS, MERS, and COVID-19 epidemics, facilitate the adoption of health policies globally, and diversify medical research by increasing international collaborations[8].

When examining international citation patterns, we identified considerable domestic preference. The nationality of the research institute accounted for 74.5% of the international citation variance. In addition, the DSCI varied 8-fold between the studied countries emphasizing that further research in citation patterns is needed. The results are in line with a study reporting that 85% of the variability of the scientific journalism coverage of 22 newspapers focused on domestic research[9]. Similar findings in citation patterns have also been observed in global science emphasizing the amplitude of the phenomenon[10]. However, we also observed international citing clusters reflecting likely similar research interests and collaborations.

We limited the scope of this study to available geographical data recorded in scientific indexing databases. Therefore, for example ethnical, socioeconomical, career-stage, and gender disparity was not examined. Similarly, the number of submissions and geographical coverage of reviewers are unavailable, while these could help to further interpret the results. While the study

examined only journals covering all medical fields, we postulate that the findings can be equally reproducible in journals specialized to distinct medical domains.

In summary, this computational audit of author affiliations in top medical journals revealed the Anglocentric dominance embodied as publication excess in relation to their accumulated citation and domestic preference by journals. Based on longitudinal data, the international representation gap in top medical journals is gradually decreasing.

## Data availability

Source data for all figures and a 100-row example of the raw data from the Clarivate Web of Science is available at https://github.com/obruck/International-Research-Impact[11]. The full raw data can be downloaded from Clarivate Web of Science, with instructions provided in the Github repository.

## Code availability

Codes are available at https://github.com/obruck/International-Research-Impact[11].

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

## Acknowledgements

The author wishes to thank Susanna Lallukka-Brück for her patience, understanding, and insightful comments. The author is grateful to Olli Dufva and to the members of the Hematoscope Lab for discussion and comments. This study was supported by research grants from the Helsinki University Hospital. Certain data included herein are derived from Clarivate Web of Science. Open access funded by Helsinki University Library.

## Author contributions

Conception and design; Collection and assembly of data; Data analysis; Manuscript writing; Manuscript editing; Data interpretation; Final approval of manuscript: O.B.

## Competing interests

O.B. declares no competing non-financial interests but the following competing financial interests: consultancy fees from Novartis, Sanofi, and Amgen, outside the submitted work; research grants from Pfizer and Gilead Sciences, outside the submitted work.
