## [Peer Review File · Communications Medicine]

Reviewers' comments:

Reviewer #1 (Remarks to the Author):

Dear Madams and Sirs,

Thank you for submitting the manuscript entitled:

The Geographical Gap in Leading Medical Journals - a Computational Audit

I was pleased to review the other manuscript regarding the gender gap with probably the same datasets. This manuscript also has some significant writing issues, both grammatical and linguistic. Unfortunately, here also, the style of the text could be more professional and suitable for publication. The quotations/references are not by the journal's required reference style, especially when quoting from an URL. The figures need some resolution, more specific legends, and descriptions of the images presented here. The tables provided must be formatted correctly; they are too crowded, the letters are not scientific and too confusing, and they should be better organized.

The study design needs to be organized, and there needs to be a clear description of the methods and how they have evaluated the journals. What are the authors presenting? The whole manuscript is abysmally organized. There are no structures whatsoever – please write the manuscript how it is supposed to be written for a journal.

Reviewer #2 (Remarks to the Author):

The article entitled “The Geographical Gap in Leading Medical Journals - a Computational Audit” is a wide-ranging descriptive assessment of we examined possible bias related to the geographical location of the conducted research to the number of publications and citations.

Overall, this paper is an interesting "audit" that uses descriptive statistics to unpack the various ways in which geography manifests in leading medical outlets. The paper finds that author affiliations in top medical journals revealed the Anglocentric dominance embodied as publication excess in relation to their accumulated citation and domestic preference by journals. However, the international representation gap in top medical journals is gradually decreasing. This is an interesting finding. The paper also examines different levels of analysis by including cities, as well. This paper could have an impact on geographic and national inequality in medical research and warrant publication but would require some non-trivial edits.

First, I would request that the paper’s introduction be rewritten to reflect the motivation of the article and the findings. Right now, the introduction curiously opens with a brief discussion of JIF and monitoring “diversity” more broadly. This doesn’t comport with the analyses and discussion. For example, there’s no discussion of or connection with medical publications. There are a lot of interesting findings here, like how publications from elite research institutes tend to accumulate fewer citations per article. This is a research paper in its own right.

Second, there needs to be a stronger justification for why only five elite journals are the focus. This is a particularly important point. The paper is not focused, as there is a misalignment between goals and methods.

Third, the Domestic Self-Citation Index is an interesting measure, but ratios can often lead to extreme results. As this is a ratio of ratios, I also request that the paper does some robustness checks to make sure the results aren't driven by small denominators/large numerator ratios for certain countries. A natural log transformation of the numerators and denominators or adding a value of 1 to numerators and denominators can help to test if this is the case.

Finally, referencing this as a "geographical gap" is a bit of awkward phrasing when compared to the more well-known "gender gap." I would recommend rethinking the title.

Reviewer #3 (Remarks to the Author):

Overall

Dr. Bruck has submitted an article highlighting geographic publication bias amongst leading medical journals. Using sophisticated statistical methods, he approaches his analysis from several different angles, including country-level, language-level and municipality level. He also introduces two ostensibly novel analytic tools, the IRI and the DSCI.

Overall, while I believe that this article has value, it proves very challenging to read. The density of statistical analysis, the disordered organization (including what appears to be missing methodology, such as explanation of the DSCI) and excessive, confusing figures all suggest that the manuscript requires significant revision prior to acceptance for publication. My specific comments are detailed below; however, I would urge the author to emphasize the most relevant aspects of his analysis that support the argument of publication biases. Not every detail requires inclusion and not every level of analysis necessarily supports his conclusion. No doubt, a lot of thoughtful work has gone into writing this manuscript and I congratulate the author on his work.

Abstract

-No specific comments.

Introduction

-It is unclear how the opening paragraph discussing Journal Impact Factor is relevant to the remainder of the introduction, or to the remainder of the manuscript. Consequently, I would eliminate this paragraph, or, if desired to keep it, suggest making it more relevant to the primary purpose of the manuscript.

Main

-Lines 94-96: The journals (NEJM, JAMA, etc.) should be spelled out in their entirety upon first use.

-Lines 97-98: "Each article included 1.2 mean affiliations [range 1-15] equaling to 10,73297 total unique entries, of which 100.0% (n=10,730) could be geolocated." As written, I'm having difficulty understanding what this means. Is it that 2 could not be located whatsoever, either on a map, or through an Internet search? Upon reading the methods, the above statement becomes more clear. The article would be much easier to understand if the methods section followed the introduction and preceded the findings.

-Lines 113-14: "We identified 19/32 countries with over 10% publication excess or deficit when normalized by their citation frequencies." Can the authors better explain how they determined "excess

or deficit"? In other words, how can a country be expected to have a certain number of publications (and, therefore, have excess or deficit)?

-Lines 115-120: I am confused by the difference between "based on the relative number of articles" and "based on the absolute number of articles." How does the author define relative number of articles when he already has an exact count? What is the relevance of either of these calculations? Is he stating that Bangladesh is overrepresented, which seems unlikely, or the U.S., which seems more likely?

-Lines 175-193: I am unclear as to the relevance of the "most productive institutes." While perhaps interesting to some readers, this line of statistical analysis does not strike me as lending to the argument of geographic publication bias.

-Lines 216-18: "To highlight the proportion of self-citations compared to the proportion of citations originating from abroad, we introduce the domestic self-citation index (DSCI, see Methods)." In reading the methods, I do not see any discussion of the DSCI. Therefore, this paragraph and the ensuing one are very confusion. Please clarify.

Discussion

-Line 242: "socioeconomical" is mis-spelled.

-Lines 241-45: The second paragraph of this section appears to discuss study limitations. This part of the discussion would be better toward the end of the section. The beginning of the discussion should start with a very brief (1-2 line) summary of the major/pertinent findings of the study.

-Lines 257-62: Again, I'm not fully appreciating the relevance of publication frequency by institution. Why is this a bad thing? This paragraph essentially re-states the findings from the previous section and should be shortened significantly. The author would do better by citing relevant literature supporting the argument that institutional representation creates bias.

-Lines 266-68: "The publication fees require considerable research funding, which are primarily available in high-income countries making scientific publishing an unintended luxury." To my knowledge, neither NEJM, nor JAMA, nor Lancet require publication fees. If this is the case, the author's argument is not supported.

Methods

-Can the authors explain the utility of the IRI? What does it demonstrate and how can it be used to identify geographic publication bias?

-Where is discussion of the DSCI?

-Why do the methods come after the results/discussion?

Tables/Figures

-General comment: many of the figures are very busy, making them difficult to read. Additionally, I strongly recommend cutting back on the number of figures for simplicity. Not all of them are necessary.

-For example, what do the different rows/columns and/or colors represent in figure 1d? Presumably,

each individual graph and/or color represents one country. Can the reader reliably differentiate between the blue of Brazil and the ever-so-slightly darker blue of Finland? Why is it necessary to graph each country?

-Figure 2c: What is the relevance of capital vs. non-capital cities (similar question for the body of the manuscript)? How does this increase bias?

-Figure 3b: This is a very confusion graphic. I am completely lost, even after reading the legend several times. What do the varying colors represent along the Y-axis (citing country)? I cannot make sense of how the colors are organized.

I thank the Reviewers for thorough assessment of the manuscript. The valuable comments helped me to edit the manuscript to ensure that the work is scientifically interesting and clear for a broad audience. Technical concerns were addressed by describing all methodological details in full and performing new analyses according to Reviewers' comments.

The main modifications are:

- Text editing to define sufficiently concepts, align the Introduction section and clarify the manuscript.
- Figures: Moving Fig. 1d and 2c to Supplementary Figures and simplify both Fig. 3b and its legend.
- New data: Supplementary Fig. 5 and Supplementary Table 1 to confirm the robustness of the Domestic Self-Citation Index.

Replies to Reviewer comments are highlighted in yellow. Here, I have marked removed words with ~~strikethroughs~~ and highlighted added words in red.

I have located the edits in the manuscript to the file named "Bruck_Manuscript_CommsMed_tracked_changes.docx". There, I have marked removed words with ~~strikethroughs~~ and highlighted added words in yellow.

Reviewers' comments

Reviewer #1 (Remarks to the Author):

Dear Madams and Sirs,

Thank you for submitting the manuscript entitled:

The Geographical Gap in Leading Medical Journals - a Computational Audit

I was pleased to review the other manuscript regarding the gender gap with probably the same datasets. This manuscript also has some significant writing issues, both grammatical and linguistic.

I sincerely appreciate the Reviewer's time and effort to evaluate both manuscripts. Each concern has been carefully addressed to ensure the manuscript meets high standards of scientific publication.

I have taken additional care to improve the readability of the text. For example, the Introduction section (L84-130) has been rewritten to better align with the manuscript's main focus. In addition, I have corrected grammatical and linguistic mistakes and better defined novel concepts (L164-180, 217-222, 418-419).

The figures need some resolution, more specific legends, and descriptions of the images presented here. The tables provided must be formatted correctly; they are too crowded, the letters are not scientific and too confusing, and they should be better organized.

To address concerns about the figures, I have made sure that they have sufficient resolution (300 dpi) for better visualization. I have also simplified and resized all figures to ensure optimal magnification. In addition, the legends of Fig. 3c has been extended to clarify the analysis. The revised figures aim to enhance the reader's understanding of the data and contribute to the overall clarity of the paper.

In response to the Reviewer's feedback, I have clarified the Table 1 by removing one column, removed colors, and adjusted the font style and size to ensure a more scientific and less confusing presentation.

Unfortunately, here also, the style of the text could be more professional and suitable for publication. The quotations/references are not by the journal's required reference style, especially when quoting from an URL.

The study design needs to be organized, and there needs to be a clear description of the methods and how they have evaluated the journals. What are the authors presenting? The whole manuscript is abysmally organized. There are no structures whatsoever – please write the manuscript how it is supposed to be written for a journal.

I apologize for the confusing manuscript structuring (e.g., Abstract), arrangement (e.g., Results section before Methods section) and selection of reference style (e.g., URL citation). To facilitate sending manuscripts for peer-review, many scientific journals, including Communications Medicine, do not require strict adherence to

journal formatting requirements upon initial submission (<https://www.nature.com/commsmed/submit/submission-guidelines#submission-policies>). The formatting has now been revised to match the journal's guidelines.

Moreover, I have re-evaluated the manuscript's study design, clearly outlining the methods employed and explaining why and how they were applied to assess the journals (L104-130, L135-137 and L164-180). To improve the robustness of the methods, I have provided additional data (Supplementary Table 1 and Supplementary Fig. 5). I have also ensured that the objectives and contributions of our research are explicitly stated (L121-130), providing a more coherent and organized manuscript overall.

Collectively, I have taken into account all comments raised by Reviewer 1 and made the necessary revisions to address each concern raised. I would like to thank the valuable feedback, which has helped to enhance the quality of the manuscript.

Reviewer #2 (Remarks to the Author):

The article entitled “The Geographical Gap in Leading Medical Journals - a Computational Audit” is a wide-ranging descriptive assessment of we examined possible bias related to the geographical location of the conducted research to the number of publications and citations.

Overall, this paper is an interesting "audit" that uses descriptive statistics to unpack the various ways in which geography manifests in leading medical outlets. The paper finds that author affiliations in top medical journals revealed the Anglocentric dominance embodied as publication excess in relation to their accumulated citation and domestic preference by journals. However, the international representation gap in top medical journals is gradually decreasing. This is an interesting finding. The paper also examines different levels of analysis by including cities, as well. This paper could have an impact on geographic and national inequality in medical research and warrant publication but would require some non-trivial edits.

First, I would request that the paper’s introduction be rewritten to reflect the motivation of the article and the findings. Right now, the introduction curiously opens with a brief discussion of JIF and monitoring “diversity” more broadly. This doesn’t comport with the analyses and discussion. For example, there’s no discussion of or connection with medical publications. There are a lot of interesting findings here, like how publications from elite research institutes tend to accumulate fewer citations per article. This is a research paper in its own right.

I am sincerely thankful for the Reviewer’s thoughtful comments and suggestions. I agree with the observation that the Introduction was disconnected from the further analyses and discussion. Due to the lack of retrospective analyses in geographical representation, especially in medical research, I have rewritten the Introduction section to focus on the general relevance of the study and motivate the use of journals with high JIFs to study geographical bias.

The edits are found in L83-130.

Second, there needs to be a stronger justification for why only five elite journals are the focus. This is a particularly important point. The paper is not focused, as there is a misalignment between goals and methods.

I agree on that clear justifications are needed for selecting only five elite journals. These explanations have been clarified in the Introduction section in L111-118

“Medical journals with high journal impact factors (JIF) are particularly interesting to study. First, the editorial and review processes of these journals are exigent to guarantee publication of cutting-edge research. Therefore, we hypothesized that prominent medical journals should not differ by the geographical representation of their publications. Second, JIFs reflect averaged citation patterns of one journal at a certain time period. We hypothesized that citation rates of articles published in prominent medical journals should not differ depending on the geographical location of the conducted research.”

Methods section L135-137

“We selected the five journals publishing mainly original articles in all fields of medicine and ranked highest in the Journal Citation Reports 2022 JIF in the field of medicine to exclude possible bias related to specialized medical fields.”

Discussion section L421-423

“While the study examined only journals covering all medical fields, we postulate that the findings can be equally reproducible in journals specialized to distinct medical domains.”

The Reviewer also raised a comment, whether the manuscript could be better focused. The manuscript describes representation at three geographical levels (continental, international and municipal) and its temporal evolution during 2010-2019. The main metric was the IRI reflecting the average frequency of citations per article. Therefore, the manuscript also interrogated possible citation patterns at the international level. I believe that these novel and sound analyses are required to make a balanced manuscript. However, some results (Fig. 1d and Fig. 2c) have been moved to Supplementary Figures to keep the main Figures clear.

Third, the Domestic Self-Citation Index is an interesting measure, but ratios can often lead to extreme results. As this is a ratio of ratios, I also request that the paper does some robustness checks to make sure the results aren't driven by small denominators/large numerator ratios for certain countries. A natural log transformation of the numerators and denominators or adding a value of 1 to numerators and denominators can help to test if this is the case.

This is an excellent statistical comment. I confirm the robustness of the Domestic Self-Citation Index by providing additional data

- (1) The range of the numerator is 3.2-37.4. The value represents the proportion (%) of citations from country X, which are citing an article from the same country.
- (2) The range of the denominator is 15.0-359.6. The value represents the proportion between the sum of citations country X receives from other countries (range 1811.0-226603.0) and the sum of citations country X cites other countries (range 1764.0-137558.0). No small/large numerator/denominator ratios were observed.
- (3) I added the value 1 to either the numerator or the denominator when calculating the Domestic Self-Citation Index. These indices correlated with the Domestic Self-Citation Index with $R=0.99$ (Spearman, $p<0.001$) and produced nearly identical results. These are provided in Supplementary Figure 5 and shown also in the next page. The Domestic Self-Citation Index section has also been clarified in L172-180 of the Methods section and L333-336 of the Results section
“To highlight the proportion of self-citations compared to the proportion of citations originating from abroad, we introduce the ~~domestic self-citation index~~ (DSCI, (see Methods “Domestic Self-Citation Index (DSCI)” and robustness tests and data in Supplementary Table 1 and Supplementary Fig. 5).”
- (4) The counts used for the Domestic Self-Citation Index are now available in Supplementary Table 1 and L172-180 of the Methods section.

Supplementary Figure 5. Tests to confirm the robustness of the Domestic Self-Citation Index. (a) Correlation plots between the original Domestic Self-Citation Index and itself after adding 1 to its numerator or (b) to its denominator. (c) Bar plot illustrating the proportion of much other countries have cited any given country in relation to how much they have cited all other countries, when adding 1 to the numerator or (d) to the denominator of the Domestic Self-Citation Index.

Finally, referencing this as a “geographical gap” is a bit of awkward phrasing when compared to the more well-known “gender gap.” I would recommend rethinking the title.

I agree with the Reviewer’s perspective and have replaced the title to "**Uncovering the Geographic Disparity in Prominent Medical Journals**" to make a distinction and better capture the focus of the manuscript.

Lastly, I would like to express my gratitude for the Reviewer for their valuable feedback and input, which has improved the clarity and relevance of the manuscript. I hope these revisions sufficiently address all concerns and make the paper more engaging to the readers.

Reviewer #3 (Remarks to the Author):

Overall

Dr. Bruck has submitted an article highlighting geographic publication bias amongst leading medical journals. Using sophisticated statistical methods, he approaches his analysis from several different angles, including country-level, language-level and municipality level. He also introduces two ostensibly novel analytic tools, the IRI and the DSCI.

Overall, while I believe that this article has value, it proves very challenging to read. The density of statistical analysis, the disordered organization (including what appears to be missing methodology, such as explanation of the DSCI) and excessive, confusing figures all suggest that the manuscript requires significant revision prior to acceptance for publication. My specific comments are detailed below; however, I would urge the author to emphasize the most relevant aspects of his analysis that support the argument of publication biases. Not every detail requires inclusion and not every level of analysis necessarily supports his conclusion. No doubt, a lot of thoughtful work has gone into writing this manuscript and I congratulate the author on his work.

I am sincerely thankful for the constructive feedback and valuable insights provided by Reviewer 3, which have improved the clarity and organization of the manuscript.

Abstract

-No specific comments.

Introduction

-It is unclear how the opening paragraph discussing Journal Impact Factor is relevant to the remainder of the introduction, or to the remainder of the manuscript. Consequently, I would eliminate this paragraph, or, if desired to keep it, suggest making it more relevant to the primary purpose of the manuscript.

I agree with the observation that the Introduction was disconnected from the further analyses and discussion. Due to the lack of retrospective analyses in geographical representation, especially in medical research, I have rewritten the Introduction section to focus on the general relevance of the study and motivate the use of journals with high JIFs to study geographical bias.

The edits are found in L83-130.

Main

-Lines 94-96: The journals (NEJM, JAMA, etc.) should be spelled out in their entirety upon first use.

I moved the Methods before the Results according to the journals formatting guidelines, which corrected this issue. The abbreviations are spelled out in L137-139.

-Lines 97-98: "Each article included 1.2 mean affiliations [range 1-15] equaling to 10,73297 total unique entries, of which 100.0% (n=10,730) could be geolocated." As

written, I'm having difficulty understanding what this means. Is it that 2 could not be located whatsoever, either on a map, or through an Internet search? Upon reading the methods, the above statement becomes more clear. The article would be much easier to understand if the methods section followed the introduction and preceded the findings.

I apologize for the confusion, and moved the Methods before the Results according to the journals formatting guidelines. Two articles included institute affiliations, which could not be geolocated with Google's search engine. This has been further clarified in the Methods section L160-161

"In total, only two institutes could not be geolocated due to insufficient information."

-Lines 113-14: "We identified 19/32 countries with over 10% publication excess or deficit when normalized by their citation frequencies." Can the authors better explain how they determined "excess or deficit"? In other words, how can a country be expected to have a certain number of publications (and, therefore, have excess or deficit)?

I thank the Reviewer for raising the important comment. The five medical journals were selected due to their high journal impact factors (JIFs). While it is difficult to predict the future citation count of any publication, there should not be notable differences by the country where the research has been conducted. In other words, publications conducted in an institute located in UK or Belgium should accumulate equal citation counts per article over a period of 10 years. However, our study indicates the opposite. Publications from Belgium accumulated citations at higher pace than publications from UK. This implies that the location of the conducted research impacts the probability of an article being accepted. I termed countries with a lower probability to have a publication deficit and those with a higher probability to have a publication excess.

I have edited L212-220 in the Results section to clarify the concepts

"While it is challenging to predict the future citation count of any publication, we hypothesized that there should not be notable differences by the country where the research has been conducted. When examining the international research impact (IRI) by countries, publication count explained 96.5% of total citations corresponding to an excellent statistical correspondence (Fig. 1a). However, we also observed unexpected discrepancies. We identified 19/32 countries publishing more than 10% more (publication excess) or less (publication deficit) articles than was expected based on their accumulated citations with over 10% publication excess or deficit when normalized by their citation frequencies."

-Lines 115-120: I am confused by the difference between "based on the relative number of articles" and "based on the absolute number of articles." How does the author define relative number of articles when he already has an exact count? What is the relevance of either of these calculations? Is he stating that Bangladesh is overrepresented, which seems unlikely, or the U.S., which seems more likely?

Relative and absolute number of articles have distinct significance. While the absolute number of articles is easier to understand, it is affected by the total number of articles a country publishes. However, the relative number of articles is defined by the publication excess/deficit of a country in relation to the total publication count of

that country, which is not affected by the frequency of articles a country publishes but does not necessarily name the countries, which have the highest publication excess/deficit (USA, UK).

In summary, these measures mediate distinct messages. Given the low number of articles in Bangladesh, I agree that it is difficult to make conclusions concerning its publication excess/deficit. This section has been rearranged and revised in L220-228: ~~“Based on the relative number of articles, Spain published 78.7% (n=76) and Brazil 62.1% (n=26) fewer and Bangladesh 54.3% (n=6) too many articles than expected. Based on the absolute number of articles, 1,174 (23.0%) and 410 (24.3%) excess articles were published by corresponding authors affiliated to an organization located in the USA and UK, respectively. Together, USA and UK accounted for 85.7% of all articles published in excess (n=1,847). When normalizing the absolute number of excess/deficit articles to each country’s total article count, Spain published 78.7% (n=76) and Brazil 62.1% (n=26) fewer articles than expected.”~~

-Lines 175-193: I am unclear as to the relevance of the “most productive institutes.” While perhaps interesting to some readers, this line of statistical analysis does not strike me as lending to the argument of geographic publication bias.

The bibliometric data permits examining the geographical distribution of articles published in high-impact medical journals at the continental, international and municipal levels. The varying publication excess/deficit at the municipal level implies that the root reason of the geographical bias is likely prestigious research institutes. Moreover, the temporal analysis indicated that the difference was diminishing both at municipal and international levels.

These findings are novel and central to understand both the reasons and the likely future development of geographical representation. However, to simplify the section, L272-309 have been shortened, some central analyses removed, and the text thoroughly revised.

“Next, we examined municipal IRIs by including only cities with ≥ 10 publications in 10 years retaining 94.0% of the number of publications (n=9,327). Publication number explained 83.6% of the variability in total accumulated citations (Fig. 2b). Multiple metropolitan cities known for their established research institutes were overrepresented in terms of publication frequencies (Fig. 2b). ~~Unexpectedly, the residuals of the linear regression model increased by publication number (corr 0.37, $p < 0.001$) manifesting as Unexpectedly, -21 out of the 25 most productive cities accumulating ed less citations than predicted with linear regression suggesting selection bias.~~ Cities with the highest absolute publication excess included Boston, USA (n=728 excess articles), London, UK (n=285) and Oxford, UK (n=76). On the contrary, cities with the highest publication deficit included Seattle, USA (n=76), Houston, USA (n=27) and Los Angeles, USA (n=6). ~~However, capital cities generally associated with elevated population density and administrative and business activity, did not accumulate more citations compared to non-capital cities (Supplementary Fig. 2).~~

While the publication count decreased in the USA and UK during the 10-year follow-up, similar patterns were not as evident when examining the **cities publishing most articles 25 most productive institutes** (Supplementary Fig. 3a). ~~The publishing dynamics varied with~~ Some cities **published ing** gradually more (New York, Stanford, Philadelphia), some less (London, Baltimore, Ann Arbor), but for most the direction was inconclusive. As in the national level, the **average** yearly citations-per-article **measure increased in average also at the institute level and the slope tendency varied more in cities with fewer publications** (Supplementary Fig. 3b). Houston stood out with higher citations/publication across follow-up, whereas publications originating from Boston, Toronto, and Copenhagen accumulated fewer citations compared to other institutes.

~~The absolute number of publications correlated moderately with population-normalized publication frequencies (corr 0.41, p<0.001). However, The residuals of the linear regression model between publication number and citations increased (corr 0.37, p<0.001) by publication number (corr 0.37, p<0.001). This implies signifying that articles originating from established research organizations tended to accumulate fewer citations than statistically would be expected. Finally, institutes from capital cities did not accumulate more citations per article compared to non-capital cities indicating that the governmental status or population of the city are poor predictors of IRI (Fig. 2c and Supplementary Fig. 2)."~~

-Lines 216-18: "To highlight the proportion of self-citations compared to the proportion of citations originating from abroad, we introduce the domestic self-citation index (DSCI, see Methods)." In reading the methods, I do not see any discussion of the DSCI. Therefore, this paragraph and the ensuing one are very confusion. Please clarify.

Thank you for the comment. To distinguish the description of the DSCI, I have separated the "Indices" chapter at the Methods section into "*The international research impact (IRI) index*" and "*The Domestic Self-Citation Index (DSCI)*" chapters, expanded the description and provided additional data. In addition, the reference to the Methods part has been ameliorated.

L172-180

"The Domestic Self-Citation Index (DSCI)

To examine the likelihood of citing publications from the same country rather from another country, we introduce DSCI. Here, we normalize national self-citations by the citation count attributed by other countries ~~defined the Domestic Self-Citation Index (DSCI) as:~~

$$\frac{\# \text{ Citations from country } \alpha \text{ to country } \alpha}{\# \text{ Total citations from country } \alpha} \\ \frac{\# \text{ Citations to country } \alpha \text{ from all countries except country } \alpha}{\# \text{ Citations in total to all countries except country } \alpha \text{ from country } \alpha}$$

where "# Citations" represents the number of citations and α any country of interest."

L333-336

"To highlight the proportion of self-citations compared to the proportion of citations originating from abroad, we introduce the **domestic self-citation index (DSCI)**, (see

Methods “Domestic Self-Citation Index (DSCI)” and robustness tests and data in Supplementary Table 1 and Supplementary Fig. 5).”

Discussion

-Line 242: “socioeconomical” is mis-spelled.

This has been corrected in L421-423.

-Lines 241-45: The second paragraph of this section appears to discuss study limitations. This part of the discussion would be better toward the end of the section. The beginning of the discussion should start with a very brief (1-2 line) summary of the major/pertinent findings of the study.

I appreciate the comment to improve reader experience.

The Discussion section’s first chapter in L353-357 has been edited

“Available bibliometric data can reveal important information on the ~~equity and geographical~~ representation in medical research and its temporal dynamics. ~~Longitudinal data permits the study of temporal trends and can be repurposed for monitoring of diversity.~~ Here, we presented geographical publication disparity in five leading medical journals based on JIFs between 2010-2019.”

The Discussion’s second chapter has been moved to the end of the section and extended

L417-423 “We limited the scope of this study to available geographical data recorded in scientific indexing databases. Therefore, for example ethnical, ~~socioeconomical~~ ~~socioeconomical~~, career-stage, and gender disparity was not examined. Similarly, the number of submissions and geographical coverage of reviewers are unavailable, while these could help to further interpret the results. ~~While the study examined only journals covering all medical fields, we postulate that the findings can be equally reproducible in journals specialized to distinct medical domains.~~”

-Lines 257-62: Again, I’m not fully appreciating the relevance of publication frequency by institution. Why is this a bad thing? This paragraph essentially re-states the findings from the previous section and should be shortened significantly. The author would do better by citing relevant literature supporting the argument that institutional representation creates bias.

I agree with the Reviewer that a city publishing more in high-impact journals than another is not an issue. However, the data indicates that certain cities publish more articles than others while these accumulate systematically fewer citations per article. The finding is unlikely to be explained by chance but indicate a bias favoring manuscripts submitted from certain institutes over others.

However, to avoid restating previous findings, I have shortened the section in L375-382.

“Second, by comparing ~~accumulated citations per article (or IRIs)~~ our analysis highlighted an undocumented discrepancy favoring notably highly productive institutes and countries. ~~The results indicate that IRI is an objective and simple measure to study geographical diversity both in medical or interdisciplinary domains and its temporal evolution. For instance, 86% of all articles published in excess based on their IRI originated from USA or UK. At the municipal level, 21/25 of the leading centers were~~

~~found to publish more articles than expected. In particular, the number of publications with a corresponding author located in Boston exceeded 60% its predicted quantity."~~

-Lines 266-68: "The publication fees require considerable research funding, which are primarily available in high-income countries making scientific publishing an unintended luxury." To my knowledge, neither NEJM, nor JAMA, nor Lancet require publication fees. If this is the case, the author's argument is not supported.

The Reviewer made an excellent point that NEJM and JAMA do not charge publication fees and, therefore, the sentence cannot be generalized to all journals. This sentence has been now removed.

Methods

-Can the authors explain the utility of the IRI? What does it demonstrate and how can it be used to identify geographic publication bias?

Thank you for the questions. IRI informs which countries publish articles accumulating most citations making it suitable for studying geographical biases. The index is very easy to calculate and simple to interpret making it ideal both for layman and researcher use. IRI can be used for a group of journals (as in this study) or for individual journals of distinct domains (cardiology, hematology, oncology etc). One interesting use-case would be to use IRI for interdisciplinary comparisons (medical vs. technical fields). In addition, the metric is objective and can be used with temporal data as in this article, making it suitable for monitoring purposes.

This have been added to the Discussion L375-382

"Second, by comparing **accumulated citations per article (or IRIs)** our analysis highlighted an undocumented discrepancy favoring notably highly productive institutes and countries. **The results indicate that IRI is an objective and simple measure to study geographical diversity both in medical or interdisciplinary domains and its temporal evolution.**"

-Where is discussion of the DSCI?

A reply to this comment is provided above. I have edited the texts L172-180 "***The Domestic Self-Citation Index (DSCI)***"

-Why do the methods come after the results/discussion?

I apologize for the confusion. The Methods have been moved now before the Results according to the journals formatting guidelines.

Tables/Figures

-General comment: many of the figures are very busy, making them difficult to read. Additionally, I strongly recommend cutting back on the number of figures for simplicity. Not all of them are necessary.

I sincerely appreciate the comment raised by the Reviewer and have now reduced and simplified main figures. See next comments.

-For example, what do the different rows/columns and/or colors represent in figure 1d? Presumably, each individual graph and/or color represents one country. Can the reader reliably differentiate between the blue of Brazil and the ever-so-slightly darker blue of Finland? Why is it necessary to graph each country?

The comment has been seriously considered and I have implemented two edits. First, as some readers are interested to understand the temporal evolution of IRI in each country, the figure has been moved to the Supplementary Fig. 1. This also simplifies the main figures. Second, the colors of the line plots have been changed to black as these did not have any significance as pointed by the Reviewer.

-Figure 2c: What is the relevance of capital vs. non-capital cities (similar question for the body of the manuscript)? How does this increase bias?

Capital cities are usually larger by area, densely populated and characterized as important administrative and business centers. Initially, I hypothesized that capital cities would be associated with higher citation counts per article. Therefore, the analysis was located next to Fig. 2b to help to understand these results.

However, I agree that this is not a central analysis for the manuscript and, therefore, it has been moved to Supplementary Fig. 2. In addition, L283-286 has been edited "However, capital cities generally associated with elevated population density and administrative and business activity, did not accumulate more citations compared to non-capital cities to non-capital cities (Supplementary Fig. 2)."

-Figure 3b: This is a very confusion graphic. I am completely lost, even after reading the legend several times. What do the varying colors represent along the Y-axis (citing country)? I cannot make sense of how the colors are organized.

I apologize for the confusion. The analysis is indeed complex at first. After thorough consideration, I suggest to maintain the analysis in the main figures as the results are novel and innovative.

The colors indicate clusters of countries which tend to cite each other more than other countries. I have removed the hierarchical tree to simplify the figure. In addition, I have further explained the figure legend L521-529.

"(b) Balloon plot illustrating the international citation patterns of articles published in leading medical journals. The balloon size reflects how much citations from a Citing country (y-axis) constitute of a Cited country's total received citations (x-axis). The balloon color reflects the absolute number of citations a Citing country (y-axis) accords a Cited Country (x-axis). ~~The balloon size reflects the proportion which countries have cited the most any given country (column-wise). The balloon color reflects the absolute number of citations any given country has received.~~ Citing countries have been grouped by hierarchical clustering into three subgroups (Euclidean distance y-axis font color)."

Lastly, I would like to warmly thank Reviewer 3 for sharing their valuable insights and helpful suggestions, which improved the quality and readability of the manuscript.

I wish to once again thank the Reviewers for thorough assessment of the manuscript and their positive comments.

The manuscript has been revisited to increase clarity. In particular, sections describing the new statistical indices and their results have been simplified.

Replies to Reviewer comments are highlighted in yellow. The edits are visible in the manuscript file named "Bruck_Manuscript_CommsMed_tracked_changes.docx". There, I have marked removed words with ~~strikethroughs~~ and highlighted added words in red.

Reviewers' comments:

Reviewer #2 (Remarks to the Author):

I want to thank the author for being open to my feedback. I think the submission is much approved and warrants publication. This will be an important contribution to the growing literature on geographic disparities in medical research citations. Best of luck!

I think the author adequately addressed the critiques in both papers from Reviewer 1. (I'll add the caveat that I can't say whether Reviewer 1 would actually be satisfied with the edits, but from my point of view the author did a decent job in both papers.)

Reviewer #3 (Remarks to the Author):

In this revision, Dr. Bruck has made extensive efforts to incorporate reviewer feedback. This version of the manuscript is much crisper, much cleaner and more clear. I applaud the author for his hard work and attention to detail.

Although this version reads much better than the original and I have an improved understanding of his statistical approach, I still find myself getting lost in the density of the text. In particular, while I get what the author is trying to do with the IRI and DSCI, I still find them, the paper as a whole--and even more so the figures--to be confusing. I found myself reading and re-reading the text to try to fully comprehend the author's explanations.

Moreover, while there is nothing wrong with introducing new statistical measures, such as the IRI and DSCI, I do worry that the author risks claiming bias using tools that have not been validated externally or internally.

Overall, I think this manuscript has value, but I get the sense that it is trying to do too many things all at once. I think simplification could go a long way to making the author's point, which is valid and an important one. Simplification might include separating the paper into two (or even three) separate manuscripts, one focusing entirely on validation of his new statistical markers of bias, the other then performing a bibliometric analysis.

Thank you for the opportunity to review this revision.

Reviewers' comments

Reviewer #2 (Remarks to the Author):

I want to thank the author for being open to my feedback. I think the submission is much approved and warrants publication. This will be an important contribution to the growing literature on geographic disparities in medical research citations. Best of luck!

Thank you very much for the kind words and your expertise throughout the evaluation of both manuscripts! I look forward to seeing how these works are received by the community.

Reviewer #3 (Remarks to the Author):

In this revision, Dr. Bruck has made extensive efforts to incorporate reviewer feedback. This version of the manuscript is much crisper, much cleaner and more clear. I applaud the author for his hard work and attention to detail.

Although this version reads much better than the original and I have an improved understanding of his statistical approach, I still find myself getting lost in the density of the text. In particular, while I get what the author is trying to do with the IRI and DSCI, I still find them, the paper as a whole--and even more so the figures--to be confusing. I found myself reading and re-reading the text to try to fully comprehend the author's explanations.

Moreover, while there is nothing wrong with introducing new statistical measures, such as the IRI and DSCI, I do worry that the author risks claiming bias using tools that have not been validated externally or internally.

Overall, I think this manuscript has value, but I get the sense that it is trying to do too many things all at once. I think simplification could go a long way to making the author's point, which is valid and an important one. Simplification might include separating the paper into two (or even three) separate manuscripts, one focusing entirely on validation of his new statistical markers of bias, the other then performing a bibliometric analysis.

Thank you for the opportunity to review this revision.

Thank you very much for the encouraging comments and taking the time to share your insights.

The main purpose of the manuscript is to evaluate geographical bias. The two new indices are introduced to provide statistical measures of bias that have not been described before. Their performance is evaluated here in a well-defined but comprehensive dataset. This design both ensures that readers will become interested to read the publication and encourage them to perform further validations in other datasets. Therefore, I would suggest to not split it into two publications but focus on simplifying the content.

I take the Reviewer's comment of possible confusion seriously. The description of the new statistical indices has to be clear to ensure that these are understood and re-used in future studies.

For this purpose, I have further clarified the assumptions and rationale for developing the indices (Methods and Results section) and the content of the DSCI equation (Methods section). To increase readability of the manuscript I have shortened multiple sentences.

Most figures are composed of barplots, scatter plots, and lineplots, which are generic in scientific publications. I have edited the legends of Fig. 1 and 2 and clarified the purpose of each analysis in the manuscript text where the results are presented. Although I acknowledge Fig. 3B to be more uncommon in the field, after careful

consideration I would emphasize it to be the most conclusive visualization to examine the multiple aspects of bias in citations patterns. Therefore, I have considerably edited this part of the manuscript to increase clarity and prevent confusions.

I would like to thank the Reviewer once again for their valuable insights, which helped to improve the communication of the manuscript. Collectively, all comments have been addressed and I hope that the manuscript is suitable for publication.